# Viscoelasticity of Liposomal Dispersions

**DOI:** 10.3390/nano13162340

**Published:** 2023-08-15

**Authors:** Lívia Budai, Marianna Budai, Zsófia Edit Fülöpné Pápay, Petra Szalkai, Noémi Anna Niczinger, Shosho Kijima, Kenji Sugibayashi, István Antal, Nikolett Kállai-Szabó

**Affiliations:** 1Department of Pharmaceutics, Semmelweis University, Hőgyes Str. 7, 1092 Budapest, Hungary; budai.livia@semmelweis.hu (L.B.); budaimarianna@gmail.com (M.B.); papay.zsofia@pharma.semmelweis-univ.hu (Z.E.F.P.); szalkai.petra@semmelweis.hu (P.S.); niczinger.noemi@semmelweis.hu (N.A.N.); 2Faculty of Pharmacy and Pharmaceutical Sciences, Josai University, 1-1 Keyakidai, Sakado 350-0295, Saitama, Japan; s.kiji.h@gmail.com (S.K.); sugib@josai.ac.jp (K.S.)

**Keywords:** liposome, viscoelasticity, oscillation rheology, nanotechnology, drug delivery system

## Abstract

Janus-faced viscoelastic gelling agents—possessing both elastic and viscous characteristics—provide materials with unique features including strengthening ability under stress and a liquid-like character with lower viscosities under relaxed conditions. The mentioned multifunctional character is manifested in several body fluids such as human tears, synovial liquids, skin tissues and mucins, endowing the fluids with a special physical resistance property that can be analyzed by dynamic oscillatory rheology. Therefore, during the development of pharmaceutical or cosmetical formulations—with the intention of mimicking the physiological conditions—rheological studies on viscoelasticity are strongly recommended and the selection of viscoelastic preparations is highlighted. In our study, we aimed to determine the viscoelasticity of various liposomal dispersions. We intended to evaluate the impact of lipid concentration, the presence of cholesterol or 1,2-distearoyl-sn-glycero-3-phosphocholine (DSPC) and the gelling agents polyvinyl alcohol (PVA) and hydroxyethylcellulose (HEC) on the viscoelasticity of vesicular systems. Furthermore, the effect of two model drugs (phenyl salicylate and caffeine) on the viscoelastic behavior of liposomal systems was studied. Based on our measurements, the oscillation rheological properties of the liposomal formulations were influenced both by the composition and the lamellarity/size of the lipid vesicles.

## 1. Introduction

The viscoelastic property is a natural feature of selected biological fluids (e.g., tears, synovial fluids). This kind of rheological character may play an important role in the drug development process, too, especially if the purpose is to mimic the properties of biologic systems. Thus, the viscoelastic behaviour of a pharmaceutical dosage form may contribute to its better tolerability. Since several body fluids—as mentioned above—exhibit viscoelastic behavior, this type of rheological property is particularly important, e.g., in the formulation of ophthalmic products [1,2], intraarticular preparations [3,4] and dermal fillers [5,6,7].

Regarding the particular rheological properties of tears, it is worth noting that, due to the viscoelastic property, tears can exhibit two types of properties, namely a predominantly viscous character at rest and a mainly elastic character under stress. When there is no blinking or other stress applied on the tear fluid, the tears act like a viscous fluid that coats the surface of the cornea, preventing it from drying out. When stress is applied in the form of blinking, the movements of the eyelids represent stress exerted on the tear fluid, which becomes more elastic, thus resisting the wiping force of the eyelids. On this basis, artificial tears optimally possess similar rheological properties to human tears, provided by the soluble mucous glycoproteins present in physiological tears [8,9].

Similarly, it is favorable if the rheological properties of intraarticular drug delivery systems mimic the physiological behavior of the synovial fluid, namely, viscoelastic characteristics. In the case of synovial fluid, the viscoelastic property ensures that the bones in joints can move smoothly and painlessly. In a state of relaxation, when no stress is applied, the synovial fluid has a predominantly viscous property, which ensures the easy gliding of the cartilage surfaces over each another in the joints. When the joint is subjected to a heavy stress (e.g., jumping), the synovial fluid becomes more elastic to prevent the bones from contacting and colliding [10,11,12].

Thus, the viscoelastic property ensures the appropriate physiological functioning of the biological liquids mentioned. Therefore, during the formulation of drugs, the intention of ensuring a viscoelastic character may play an important role, especially in the case of ophthalmic preparations and intraarticular injections. Besides offering beneficial rheological properties, the use of viscoelastic dosage forms can offer therapeutic advantages by providing drug retention while allowing easy spreading of the formulation [8,13,14,15,16,17]. 

Nowadays, liposomes as nanosized drug delivery systems are very popular. They are able to provide sustained drug release and enhance the permeation of the encapsulated active ingredient [18,19,20,21]. Therefore, improved bioavailability, enhanced therapeutic effect and higher patient adherence can be achieved with their application [22,23,24]. Isotonic and isohydric liposomal dispersions—also without presence of active ingredients—can be successfully applied in selected cases, e.g., as artificial tears for dry eye syndrome [25,26,27]. In other cases, according to the needs, the encapsulation of appropriate drug molecules can be necessary [28,29,30,31,32]. On the basis of their physicochemical properties, active ingredients can be encapsulated into the aqueous phase of liposomes or can be located between the apolar fatty acids or bound to the polar lipid heads of the bilayer [33,34,35,36]. If cholesterol is also present, it is incorporated between the lipophilic fatty acids of the liposomal bilayer [37].

In both cases—for empty and drug-loaded liposomes—the examination of rheological properties (such as viscoelasticity) is of great importance. If the liposomal dispersion possesses the viscoelastic property, it may provide additional benefits to its therapeutic effect. Therefore, understanding and knowing the factors influencing viscoelasticity in the case of various dosage forms—among others, liposomal-based drug delivery systems—is crucial for formulation development aspects [31,32].

Our purpose was to investigate and summarize the factors influencing the viscoelastic nature of liposomal dispersions. The present study examines and summarizes the factors affecting the viscoelastic properties of liposomal dispersions. We evaluated the effect of cholesterol and 1,2-distearoyl-sn-glycero-3-phosphocholine (DSPC) in soy lecithin-based liposomes. Furthermore, we studied the influence of two model drugs (phenyl salicylate and caffeine) and selected auxiliary materials (buffer salts, gel-forming materials in the aqueous phase) on the rheological properties of liposomal vesicles with different lamellarity using dynamic oscillatory measurements. 

## 2. Materials and Methods

### 2.1. Materials

L-α-Phosphatidylcholine from soybean (LEC), cholesterol (CHOL) and 1,2-Distearoyl-sn-glycero-3-phosphocholine (DSPC) were purchased from Sigma-Aldrich Ltd. (Budapest, Hungary). All the excipients used in our experiments (hydroxyethyl-cellulose (HEC), phenyl salicylate, caffeine, sodium dihydrogen phosphate, sodium monohydrogen phosphate, polyvinyl alcohol (PVA)) were from Hungaropharma Ltd. (Budapest, Hungary). Deionized ultrafiltered water was produced by Milli-Q system (Millipore Inc., Budapest, Hungary). Polycarbonate filters with a pore size of 200 nm applied for extrusion were purchased from Schleicher & Schuell GmbH. (Dassel, Germany). Microcon YM-10 centrifugal filters with a cut-off value of 10 kDa were purchased from Millipore Inc. (Budapest, Hungary). 

### 2.2. Preparation of Liposomal Dispersions

Multilamellar vesicles (MLV) were formulated using the thin-layer hydration technique. A phospholipid stock solution of 20 mg/mL phospholipid dissolved in absolute ethanol was taken to the appropriate volume and the ethanolic solvent was evaporated applying nitrogen stream. After the evaporation, the samples were kept in a vacuum desiccator for at least 24 h. The hydration of the lipid films occurred at 60 °C, which is higher than the main phase transition temperature of the applied DSPC. Liposomal samples for oscillatory rheological and size distribution measurements were hydrated with distilled water, phosphate buffer (pH = 7.4) or hydrogels (0.5%). The final concentrations of the lipids were 7.0, 10.0, 12.0 and 15.0 mg/mL (Table 1). 

In the case of the phenyl salicylate-containing MLV samples, the phenyl salicylate was co-solved with LEC before film-preparation and then hydrated with distilled water. The LEC concentration was 10 mg/mL, the molar ratio of phenyl salicylate to LEC was ~1 to 10. For the caffeine-loaded MLV samples, the LEC films were hydrated with an aqueous caffeine solution whose caffeine concentration was 0.3 mg/mL. The LEC concentration was 10 mg/mL. Thus, the molar ratio of caffeine to LEC was approximately 1 to 10. Reduced-size vesicles (RSVs) were prepared from the LEC MLVs with 10 mg/mL lipid concentration by extrusion (Figure 1). The liposomal vesicles were extruded at 40 °C applying 41 extrusion cycles through polycarbonate membrane filters with a pore size of 200 nm.

### 2.3. Determination of the Encapsulation Efficiency

To determine the encapsulation efficiency, 400 µL of freshly prepared MLV samples containing phenyl salicylate or caffeine were centrifuged in an Eppendorf centrifuge (11,000× *g*; 20 min) through centrifugal filter devices with a cut-off value of 10 kDa. The concentration of phenyl salicylate or caffeine in the filtrate, representing the amount of non-encapsulated fractions, was determined by spectrophotometry (Unicam 2 UV/VIS spectrophotometer) at the wavelengths of 308.9 and 273.6 nm, respectively.

### 2.4. Rheological Measurements

Rheological experiments were performed by a Kinexus Pro^+^ rheometer (Malvern Instruments Ltd., Malvern, UK). The software of rSpace for Kinexus Pro 1.3. (Netzsch, Bayern, Germany) was used for registration of the measured data. A cone-and-plate geometry system was applied, ensuring a 0.15 mm gap for the sample. All measurements were performed at room temperature (25.0 °C). The temperature was controlled by the Peltier system of the rheometer. Thixotropy tests were performed using three-phase measurement tests. A shear rate of 0.1 s^−1^ was used in the first and third phase of the tests, while a shear rate of 100 s^−1^ was chosen for the second phase. The first and second test phases each lasted 30 s; the third one lasted at least 200 s. To determine the linear viscoelastic region (LVR), amplitude sweep tests were performed, applying a shear strain range of 0.1–100%. The amplitude sweep tests were followed by frequency sweep tests in order to detect the viscoelastic cross-over point, if present in the sample. A frequency range of 0.1–10 Hz was used, corresponding to an angular frequency range of 0.63–63 rad/s. During the entire measurement process, the sample was sealed with a stainless steel cover to prevent evaporation and create stable conditions for reproducible measurements. The rheological experiments were carried out three times for each sample, the mean values of the measurements are shown in the graphs.

### 2.5. Size Distribution Measurements

Size distributions of the liposomal samples were carried out with a Mastersizer 2000™ (Malvern Instruments Ltd., UK) in the dispersion unit Hydro SM (1500 rpm). Triplicate samples were analysed and the measurements were carried out within 5 min. Size distribution measurements of the extruded liposomes were carried out with the nanosizer Nano-ZS (Malvern Instruments Ltd., UK).

## 3. Results

### 3.1. Thixotropic Measurements

All samples examined were found to be thixotropic in the viscometric thixotropy tests. The thixotropic test of the 100% soy lecithin liposomal dispersion in HEC gel—as a representative example—is plotted in Appendix A.

### 3.2. Dynamic Oscillatory Measurements

All the dynamic oscillatory measurements were carried out in the linear viscoelastic region (LVR) at a strain amplitude of 60%, under which the examined samples do not suffer irreversible structural decomposition. A representative amplitude sweep test is shown in the graph (Figure 2) of the sample LEC-DSPC (90/10) MLV (10.0 mg/mL); more amplitude sweep tests can be found in the Appendix A.

Frequency sweeps were conducted in order to investigate the viscoelastic behaviour of the prepared liposomal compositions under increasing oscillatory angular frequency. The measured parameters such as phase angle δ (°), loss modulus or viscous modulus G″ (Pa) and storage modulus or elastic modulus G′ (Pa) varied with the angular frequency ω (rad/s) employed [38,39,40]. 

The viscoelastic transition point can be easily and clearly detected as it manifests in a cross-over point of the G′ and G″. Furthermore, at the viscoelastic transition point, the phase angle (δ) equals 45°. Viscoelastic gels present a phase angle (δ) of 45° at the cross-over point where G′ = G″ (Table 2) [38,41,42]. 

The cross-over point assigns the transition of viscoelastic liquids if the transition occurs from viscous-type behaviour into elastic-type behaviour [38,43]. However, viscoelastic solids show the transition from elastic-type behavior into fluid-like materials. Elastic behaviour is present if the material returns to its original shape when the applied force is removed, and viscous- or liquid-like behaviour is present if the material does not regain its original shape due to extensive deformation even after the applied force is removed [44,45,46,47].

### 3.3. Effect of Lamellarity and Size of Liposomes on Their Viscoelastic Property

Independently of the LEC concentration (7.0, 10.0, 12.0 and 15.0 mg/mL), viscoelastic characteristics can be observed for the liposomal preparations. The viscoelastic cross-over point can be detected at 0.11–0.14 Pa moduli range and at 26.0–30.0 rad/s angular frequency. It can be concluded that the concentration of lecithin does not have a significant impact on viscoelastic behaviour. For the examined LEC concentrations, the viscous properties (G″) dominate at low frequencies, while the elastic ones (G′) dominate at higher frequencies, as shown for the MLV sample with 10.0 mg/mL LEC concentration (Figure 3a, Table 2 and Table 3). On this basis, the prepared soy lecithin liposomal dispersions can be regarded as viscoelastic systems.

As a result of the extrusion process, the average hydrodynamic diameter of liposomal vesicles was reduced to about 300 nm (Figure 4).

According to our results for the LEC RSV (10.0 mg/mL) samples, the cross-over point using the frequency sweep measurement cannot be detected, allowing us to conclude that size-reduced LEC liposomal dispersion—in contrast to the multilamellar ones—does not have viscoelastic property (Figure 3b, Table 2). 

### 3.4. Effect of Lipid Composition on the Viscoelastic Property of Liposomal Dispersions

The formulation of MLVs with cholesterol (30%) or DSPC (10%)—instead of pure LEC liposomes—does not abolish the viscoelastic characteristics. The viscoelastic cross-over point of the cholesterol-containing liposomes can be detected at 0.20–0.21 Pa moduli range and at 39.0–40.0 rad/s angular frequency (Figure 5a, Table 3).

The DSPC content of the liposomal formulations does not alter significantly the oscillation rheological parameters of the pure LEC MLVs. For the DSPC-containing LEC liposomes, the viscoelastic cross-over point can be detected at 0.09 Pa modulus and at 31.5–33.5 rad/s angular frequency (Figure 5b, Table 3).

### 3.5. The Influence of Active Ingredient Encapsulation on the Oscillation Rheology of LEC MLVs

Phenyl salicylate- and caffeine-loaded MLVs were prepared in order to investigate the potential influence of the active ingredient on the viscoelastic parameters of the LEC liposomal dispersions. According to our results, neither the phenyl salicylate-encapsulated MLVs nor the caffeine-encapsulated ones presented viscoelastic character (Figure 6); the cross-over point could not be observed using the frequency sweep measurements.

### 3.6. Effect of Selected Excipients on the Rheological Characteristics of Liposomal Preparations

MLV samples (10.0 mg/mL) were hydrated with phosphate buffer (pH = 7.4) instead of distilled water in order to investigate the potential influence of phosphate buffer on the rheological properties of the MLVs. For the phosphate buffer-hydrated samples, the viscoelastic cross-over point was detected at 0.11–0.14 Pa moduli range and at 26.0–30.0 rad/s angular frequency, indicating that the phosphate buffer-hydrated liposomes preserved their viscoelastic nature (Table 3).

### 3.7. Effect of Liposomal Dispersion on the Rheological Properties of Hydrogels

Pure hydrogels (PVA 0.5% or HEC 0.5%) without liposomes do not demonstrate a viscoelastic character (Figure 7a and Figure 8a, Table 2). However, MLVs prepared from LEC (10.0 mg/mL) and hydrated with either PVA (0.5%) or HEC (0.5%) gel, the so-called liposomal hydrogel formulations, do show viscoelastic properties. Thus, originally non-viscoelastic gels can be transformed into viscoelastic ones with co-formulation with liposomes. The viscoelastic cross-over point could be detected at 0.13–0.16 Pa moduli range and at 19.0–20.0 rad/s angular frequency in the case of the PVA liposomal gel (Figure 7b, Table 3). For the HEC liposomal gel, the viscoelastic cross-over point was observed at 0.40–0.48 Pa moduli range and at 31.0–32.0 rad/s angular frequency (Figure 8b, Table 3).

### 3.8. Viscoelastic Liquids

Viscoelastic liquids are defined as systems possessing a dominant viscous modulus at lower oscillatory frequencies, while having a dominant elastic modulus at higher frequencies. The higher viscous modulus indicates the lack of strong chemical bonds between the individual molecules of the viscoelastic liquids. In the case of all our measured viscoelastic samples, a cross-over point indicating viscoelastic liquids could be observed (Table 3). Since the lipid molecules in the liposomal bilayer are bound by physical forces rather than chemical bonds, the liposomes do not tend to create viscoelastic solid materials.

## 4. Discussion

### 4.1. Effect of Lipid Concentration, Size and Lamellarity on Liposomal Viscoelasticity

In the case of all the viscoelastic liposomal dispersions, the oscillatory measurements taken by varying the frequency indicated that the viscous modulus (G″) was higher than the elastic modulus (G′) at lower frequencies than the cross-over point, reflecting that the samples belong to the category of so-called viscoelastic liquids. The observed viscoelastic liquid behavior does not show dependence on the concentration of LEC lipids in the concentration range between 7.0 and 15.0 mg/mL. No significant alteration was found in the viscoelastic “fingerprints” measured by the frequency sweeps in the MLV samples with various concentrations. Our results indicate that the size and/or lamellarity of the liposomal samples can have a pronounced effect on the presence or absence of viscoelasticity. Vesicles with an average hydrodynamic diameter of ~300 nm presented no cross-over point of viscoelasticity in the frequency sweep tests. In contrast, the MLVs with an average hydrodynamic diameter of 10,000 nm showed a viscoelastic cross-over point (Figure 3 and Figure 4, Table 2). It can be concluded that liposome size has an important impact on the viscoelastic properties of liposomal systems. The viscoelastic property of a phospholipid bilayer with resistance to stretching and bending was earlier proven by Melzak K.A. et al. using acoustic sensors [48]. According to our research, the multilamellarity of the vesicles significantly contributes to the viscoelasticity of the liposomes. Consequently, the reduction of lamellarity leads to the weakening or even to the loss of viscoelastic features, as can be observed for the RSV samples. This may be explained by the experimental fact that liposomes prepared by the thin-layer hydration technique that possess a hydrodynamic diameter of a large number of micrometers are typically multilamellar vesicles, while smaller vesicles (~100–300 nm) are mostly unilamellar vesicles. According to different diameters in the case of the MLVs and RSVs, the curvatures of the lipid bilayers are different for uni- and multilamellar vesicles, too. As a result of the different curvatures, the density and proximity of the lipid head groups and fatty acid chains is different for MLVs and RSVs, too, leading to less combined motional freedom in the case of the RSVs than in the case of the MLVs. This phenomenon may also be manifested in the weakening or potential loss of the viscoelastic characteristics detected for the extruded vesicles. Regarding the role of the concentration and size of vesicles on the rheological properties of the preparations, our findings are in contrast to the study of Mourtas et al., which concluded that composition and concentration, but not size, can be determined as influencing factors on rheology [49]. It should be mentioned that Mourtas et al. studied different types of samples, which also contained Carbopol 974 NF and natrosol. On the contrary, our experiments demonstrate the rheological effect of the concentration and size of pure liposomal aqueous dispersions [49]. Based on the observed results, it can be concluded that it is worthwhile to prepare multilamellar vesicles—which are typically larger than unilamellar ones—in order to retain the existing viscoelastic properties or impart additional ones to the formulation. On the other hand, regarding the pharmacokinetic properties of the liposomal formulations, it should be mentioned that higher lamellarity of the liposomes may lead to a slower distribution of the vesicles compared to unilamellar liposomes with a smaller diameter.

### 4.2. Effect of Lipid Composition on Viscoelasticity of Liposomal Dispersions

By incorporating 30% cholesterol or 10% DSPC into the lipid bilayer of the LEC MLVs (10.0 mg/mL), the viscoelastic character is restored and can even be increased. Due to intercalation, cholesterol is known to be able to alter the relatively fluid liposomal lipid bilayers to more rigid ones. Also, in our case, the LEC bilayer can be characterized with a relatively low main phase transition temperature (ca. ~2 °C) and as a consequence with relatively fluid membrane properties at room temperature. The addition of CHOL in 30 mol/mol % leads to a more rigid structure that also possesses viscoelastic properties. However, with more pronounced viscoelastic features than in the case of the pure LEC MLVs, the cross-over point of viscoelasticity is shifted to higher angular frequencies and to higher viscous and elastic moduli in the presence of CHOL.

According to our observations, the presence of DSPC in 10 mol/mol % does not lead to pronounced alteration of the viscoelastic features of the LEC samples. The cross-over point of viscoelasticity is not significantly shifted for the DSPC-containing liposomes in comparison to the pure LEC liposomes. 

### 4.3. Effect of Drug Encapsulation on the Viscoelasticity of Liposomes

The disappearance of the viscoelastic character that was detected in the presence of the highly lipophilic phenyl salicylate and in the presence of water-soluble caffeine is an interesting issue. Although the encapsulation efficiency of phenyl salicylate was nearly three times (86.4 ± 4.1%) as high as that of caffeine (34.2 ± 2.9%), the incorporation of caffeine into the lipid bilayer caused a significant reduction in the viscoelasticity of the vesicles (Figure 6, Table 2). It is known that the encapsulation of an active ingredient into the liposome changes the structure of the lipid bilayer. Based on our previously published electron paramagnetic resonance (EPR) and zeta potential measurements, the binding of caffeine does not alter the zeta potential nor the rotational motions of the lipid molecules around the head group region, even when the caffeine binds to the liposomal bilayer with an encapsulation efficiency of approximately ~34.2% [50]. However, based on the encapsulation, zeta potential and EPR measurements, we can accept that, while the caffeine–membrane interaction does not appear to be strong enough to modify the electrostatic interactions and the local rotational correlation time, the binding of caffeine to the liposomal membranes does occur [50]. Therefore, this kind of interaction may be responsible for the observed rheological changes and for the loss of viscoelasticity.

In the case of phenyl salicylate—as in the case of the other lipophilic photoprotector drugs—our research group measured high encapsulation efficiencies [50]. It is assumed that the lipophilic molecules deeply immerse into the lipid bilayers. As a result of the immersion, the active ingredients are snagged more or less among the fatty acid chains, which inhibit their coordinated motion, leading to a less-ordered bilayer structure with a non-viscoelastic character. In the case of the drug-to-lipid molar ratios of 1 to 10—as in the case of our study—this kind of “snagging” and molecular interaction may result in the loss of viscoelasticity.

### 4.4. Effect of Buffer as Hydrating Agent on Viscoelasticity of Liposomal Dispersions

The hydration of liposomal dispersions with phosphate buffer instead of distilled water did not cause any significant change in the viscoelasticity of MLVs; the liposomes preserved their viscoelastic features (Table 2). This observation allows us to conclude that potential changes caused in the hydration of the lipid head groups due to the presence of the ions of the buffer do not have a significant impact on the viscoelasticity of the sample. 

### 4.5. Effect of Liposomal Dispersion on the Rheological Properties of Hydrogels

In our study, the pure PVA 0.5% and HEC 0.5% gels did not show viscoelastic characteristics, presenting no viscoelastic cross-over point. This phenomenon is not surprising given that the viscoelastic behavior of gels is strongly influenced by the entanglement of their monomers. As previously noted, e.g., in the case of sodium hyaluronate-containing ophthalmic viscoelastic devices, the high molecular weight and long-chained molecules tend to intertwine and interlock with each other. The interactions of polymers lead to cohesive and more viscoelastic systems. In contrast to cohesive systems, the formulations of low molecular weight and short-chained sodium hyaluronate gels are dispersive because their short molecular chains do not interlink or become entangled easily [51,52].

The effect of added LEC liposomes on the oscillatory parameters of the PVA and HEC gel is clearly expressed due to the appearance of the viscoelastic feature. A moderate rise can be observed in the values of viscous moduli at low angular frequencies, which proves to be sufficient to transform the MLV-containing hydrogel systems into viscoelastic liquids. Both the liposomal PVA and HEC gels can be regarded as viscoelastic liquids with a dominant viscous modulus at low frequencies. It is supposed that the presence of the MLVs in the polymer systems contributes considerably to the increase of the entanglement of the gelling agent monomers. Our results for PVA and HEC hydrogels coincide with the results of Mourtas et al., which emphasize the beneficial effect of adding liposomes to hydrogels [49].

### 4.6. Physical and Chemical Interactions Resulted in Viscoelastic Liquids

While viscoelastic solids mainly consist of molecules chemically bonded and/or linked by relatively strong physical–chemical interactions, in the case of viscoelastic liquids, the interactions between the molecules are typically weaker. In the latter case, selected physical and chemical interactions or entanglements of polymer molecules can lend viscoelasticity to the systems. Our viscoelastic liposomal samples were found to be viscoelastic liquids. It can be concluded that the physical and electrochemical interactions between the lipid molecules of liposomal bilayers (e.g., apolar–apolar interactions between lipid chains, weak interactions between the lipid headgroups) result in viscoelastic liquid behaviour.

### 4.7. Summary of the Factors Influencing the Viscoelasticity of Liposomal Dispersions

In summary, it can be concluded that on the basis of our measurements the viscoelastic property—which is advantageous in the field of pharmaceutical drug delivery system development—may have been identified in the case of multilamellar vesicles independently of the concentration of LEC and of the presence of DSPC or CHOL in the lipid composition. Moreover, in the case of the PVA and HEC gels, the addition of liposomes resulted in the appearance of viscoelastic cross-over points, observations that clearly show the beneficial contribution of vesicular systems to the viscoelastic nature. However, the size and lamellarity reduction caused by the extrusion of LEC MLVs had a pronounced weakening effect on the advantageous rheological parameters of the preparations. Similarly, the addition of model drugs—phenyl salicylate or caffeine—resulted in the loss of viscoelasticity, too (Table 2).

One can conclude that the impact of drug encapsulation on viscoelasticity should be individually studied for each encapsulated drug, keeping in mind that molecular interactions between the drug and the lipids can be various depending on the physical and chemical properties of the drug. The liposomal encapsulation of a drug can occur in various ways. Lipophilic drugs are immersed and/or incorporated into the bilayer, while hydrophilic ones are typically only bound to the lipid head-groups [53,54]. As a result of this, lipophilic drugs show a far greater efficiency of encapsulation into the liposome structure than hydrophilic ones due to their deep immersion and tighter incorporation into the lipid bilayers. For purely lipophilic drugs, encapsulation efficiency seems to reach its highest value due to the lipophilic nature of the phospholipid bilayers, although for hydrophilic drugs the encapsulation efficiency remains at a very low level [52]. Therefore, it is expected that lipophilic drugs could change the viscoelastic properties of liposomes to a higher extent than hydrophilic ones. 

Regarding the medical application fields, the use of liposomes in the development of ophthalmological products may offer some advantages. The presence of liposomes may lead to increased adhesion and prolonged drug release; additionally, the presence of a liposomal system can provide viscoelastic characteristics. In order to ensure viscoelasticity, it is recommended to use multilamellar vesicles, as high lamellarity results in viscoelastic properties, as demonstrated by our results.

Similarly, the use of liposomal dispersions or gels is also strongly recommended for intraarticular injections due to the associated advantageous viscoelastic properties. Nowadays, the use of hydrogels in the formulation of intraarticular products is widespread. However, their rheological properties may be further improved by the addition of liposomes, which can endow viscoelasticity to the product in order to better mimic physiological conditions (the HEC and PVA gels were not viscoelastic, but the addition of liposomes converted them to viscoelastic gels in our experiments).

## 5. Conclusions

Our study demonstrates that the presence of empty (not drug-loaded) liposomes, depending on their lamellarity and size, may ensure viscoelastic properties by transforming non-viscoelastic systems into viscoelastic ones. On the basis of our dynamic oscillatory experiments, multilamellar vesicles prepared from soy lecithin (lipid concentration between 7.0 and 15.0 mg/mL) behave as viscoelastic liquids. However, the reduction of the lamellarity and/or size of multilamellar vesicles can lead to the disappearance of the viscoelasticity of the liposomal dispersions. Similarly, the addition of selected drug molecules may lead to the disappearance of viscoelastic characteristics. This phenomenon allows us to suppose that the mentioned drugs immerse deeply into the lipid bilayer of the liposomes, evoking disordered structures that are less resistant against deformational strain. The extent of drug immersion into the lipid bilayer of liposomes should be assessed for each drug individually. The more lipophilic the drug, the more it immerses into the lipid bilayer, altering the structure and destroying the viscoelastic properties in a more pronounced way. Worthy of note is the finding that the addition of liposomes to hydrogels may impart viscoelasticity and improved viscous and elastic moduli to previously non-viscoelastic gels. For the design and development of liposomal pharmaceutical formulations with viscoelastic characteristics, the required viscoelasticity can be more easily achieved if the lamellarity and/or the size of the vesicles is higher.

## Figures and Tables

**Figure 1 nanomaterials-13-02340-f001:**
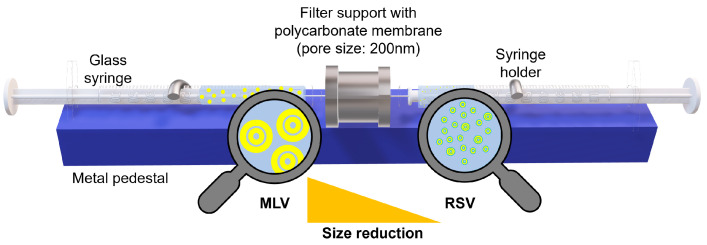
Extrusion of MLV liposomes.

**Figure 2 nanomaterials-13-02340-f002:**
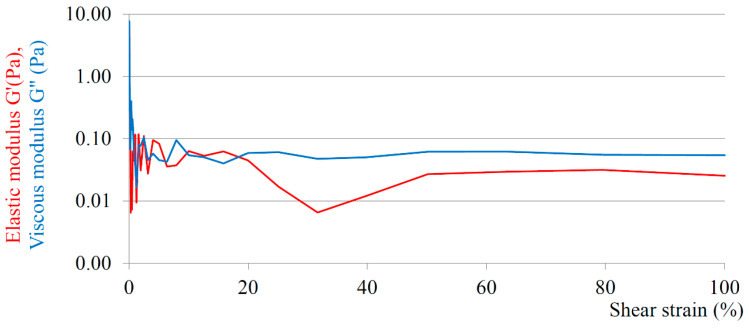
Amplitude sweep test of LEC-DSPC (90/10) MLV (10.0 mg/mL).

**Figure 3 nanomaterials-13-02340-f003:**
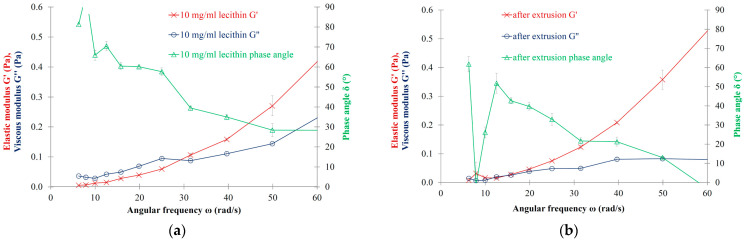
(**a**) Oscillation rheology of LEC MLV sample with 10.0 mg/mL lipid concentration; (**b**) Oscillation rheology of LEC RSV sample with 10.0 mg/mL lipid concentration.

**Figure 4 nanomaterials-13-02340-f004:**
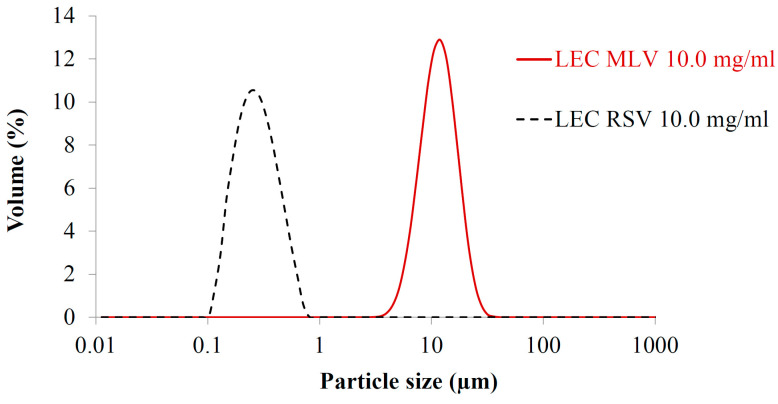
Size distribution of freshly prepared LEC MLVs and RSVs with 10.0 mg/mL lipid concentration.

**Figure 5 nanomaterials-13-02340-f005:**
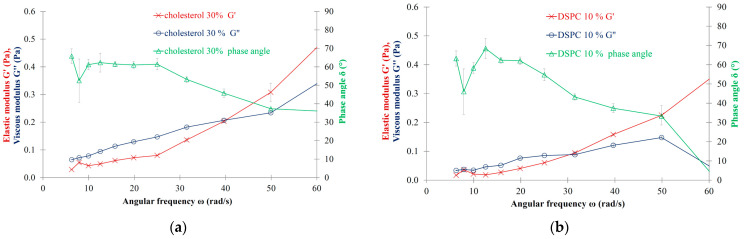
(**a**) Oscillation rheology of LEC-CHOL (70/30) MLV sample with 10.0 mg/mL lipid concentration; (**b**) Oscillation rheology of LEC-DSPC (90/10) MLV sample with 10.0 mg/mL lipid concentration.

**Figure 6 nanomaterials-13-02340-f006:**
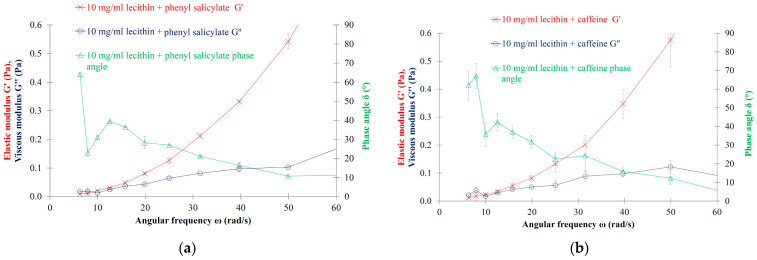
(**a**) Oscillation rheology of phenyl salicylate-loaded LEC MLVs; (**b**) Oscillation rheology of caffeine-loaded LEC MLVs.

**Figure 7 nanomaterials-13-02340-f007:**
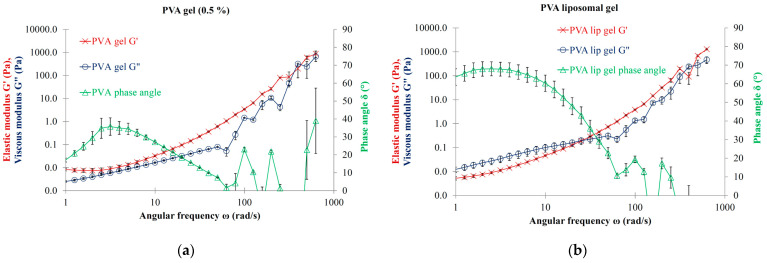
(**a**) Oscillation rheology of polyvinyl alcohol gel (0.5%); (**b**) Oscillation rheology of LEC MLVs (10.0 mg/mL lipid concentration) hydrated with polyvinyl alcohol gel (0.5%).

**Figure 8 nanomaterials-13-02340-f008:**
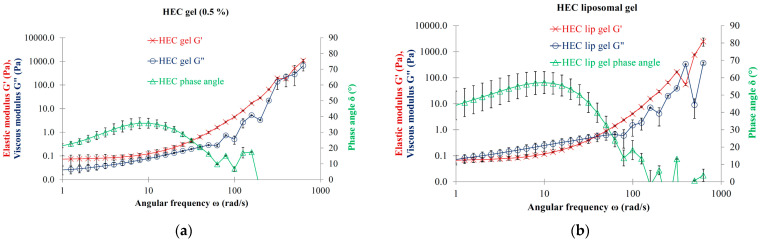
(**a**) Oscillation rheology of hydroxyethylcellulose gel (0.5%); (**b**) Oscillation rheology of LEC MLVs (10.0 mg/mL lipid concentration) hydrated with hydroxyethylcellulose gel (0.5%).

**Table 1 nanomaterials-13-02340-t001:** Composition of the prepared samples.

Name of the Sample	Composition
LEC MLV (7.0 mg/mL)	7.0 mg soy lecithin in 1 mL distilled water
LEC MLV (10.0 mg/mL)	10.0 mg soy lecithin in 1 mL distilled water
LEC MLV (12.0 mg/mL)	12.0 mg soy lecithin in 1 mL distilled water
LEC MLV (15 mg/mL)	15.0 mg soy lecithin in 1 mL distilled water
LEC RSV (10.0 mg/mL)	10.0 mg soy lecithin in 1 mL distilled water
LEC-CHOL (70/30) MLV (10.0 mg/mL)	7.0 mg soy lecithin and 3.0 mg cholesterol in 1 mL distilled water
LEC-DSPC (90/10) MLV (10.0 mg/mL)	9.0 mg soy lecithin and 1.0 mg DSPC in 1 mL distilled water
Phenyl salicylate-loaded LEC MLV (10.0 mg/mL)	0.28 mg phenyl salicylate, 10.0 mg soy lecithin in 1 mL distilled water
Caffeine-loaded LEC MLV (10.0 mg/mL)	0.3 mg caffeine, 10.0 mg soy lecithin in 1 mL distilled water
LEC MLV with phosphate buffer (pH 7.4) (10.0 mg/mL)	10.0 mg soy lecithin in 1 mL phosphate buffer
Polyvinyl alcohol gel (0.5%) without liposomes	5 mg polyvinyl alcohol in 1 mL distilled water
Hydroxyethylcellulose gel (0.5%) without liposomes	5 mg hydroxyethylcellulose in 1 mL distilled water
LEC MLV (10.0 mg/mL) hydrated with polyvinyl alcohol gel (0.5%)	10 mg soy lecithin, 5 mg polyvinyl alcohol in 1 mL distilled water
LEC MLV (10.0 mg/mL) hydrated with hydroxyethylcellulose gel (0.5%)	10 mg soy lecithin, 5 mg hydroxyethylcellulose in 1 mL distilled water

**Table 2 nanomaterials-13-02340-t002:** Division of the tested samples based on viscoelastic and non-viscoelastic characteristics.

Viscoelastic Samples	Non-Viscoelastic Samples
LEC MLV (7.0 mg/mL)	LEC RSV (10.0 mg/mL)
LEC MLV (10.0 mg/mL)	Phenyl salicylate-loaded LEC MLV (10.0 mg/mL)
LEC MLV (12.0 mg/mL)	Caffeine-loaded LEC MLV (10.0 mg/mL)
LEC MLV (15 mg/mL)	Polyvinyl alcohol gel (0.5%) without liposomes
LEC-CHOL (70/30) MLV (10.0 mg/mL)	Hydroxyethylcellulose gel (0.5%) without liposomes
LEC-DSPC (90/10) MLV (10.0 mg/mL)	
LEC MLV with phosphate buffer (pH 7.4) (10.0 mg/mL)	
LEC MLV (10.0 mg/mL) hydrated with polyvinyl alcohol gel (0.5%)	
LEC MLV (10.0 mg/mL) hydrated with hydroxyethylcellulose gel (0.5%)	

**Table 3 nanomaterials-13-02340-t003:** Rheological range of viscoelastic cross-over points for the tested samples. The characteristics of cross-over points are given in Pa for elastic and viscous moduli (G′ and G″) and in rad/s for angular frequency (ω).

Name of the Viscoelastic Sample	Cross-Over Point Characteristics (G′ and G″ in Pa)	Cross-Over Point Characteristics (ω in rad/s)
LEC MLV (7.0 mg/mL)	0.11–0.14 Pa	26.0–30.0 rad/s
LEC MLV (10.0 mg/mL)		
LEC MLV (12.0 mg/mL)		
LEC MLV (15 mg/mL)		
LEC-CHOL (70/30) MLV (10.0 mg/mL)	0.20–0.21 Pa	39.0–40.0 rad/s
LEC-DSPC (90/10) MLV (10.0 mg/mL)	0.09 Pa	31.5–33.5 rad/s
LEC MLV with phosphate buffer (pH 7.4) (10.0 mg/mL)	0.11–0.14 Pa	26.0–30.0 rad/s
LEC MLV (10.0 mg/mL) hydrated with polyvinyl alcohol gel (0.5%)	0.13–0.16 Pa	19.0–20.0 rad/s
LEC MLV (10.0 mg/mL) hydrated with hydroxyethylcellulose gel (0.5%)	0.40–0.48 Pa	31.0–32.0 rad/s

## Data Availability

Not applicable.

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
