# Peer review of "Viscoelasticity of Liposomal Dispersions"

_nanomaterials, 2023, doi:10.3390/nano13162340_

Round 1
Reviewer 1 Report
The manuscript aims to study the viscoelasticity of liposomal dispersions deepening the influence of different parameters, namely the nature and concentration of lipids, particle size and lamellarity, the presence of cholesterol and encapsulated drugs (caffeine and phenyl salicylate), as well as hydration media (i.e., salt buffer, hydroxyethyl-cellulose and polyvinyl alcohol) on the viscoelastic behavior of liposomes. The subject seems really interesting from the abstract but instead the level of the paper is instead very poor. The introduction describes neither the state of art nor the relevance of the work in the context. The aim of the paper is not clear and a critical discussion about the meaning of the results for the design of pharmaceutical and cosmetic products is missing. As an example, authors state that according to the results of the rheological studies multilamellar liposomes with a mean diameter higher than 300 nm should be preferred. However these physico-chemical properties may affect the biodistribution of the nanosystems or the skin penetration of the vesicles (when products to be applied on the skin are considered). No comments on these aspects have been done. The conclusion section is a summary of the previous statements. More importantly, the results provided in the paper do not seem to add a significant knowledge in the field. Previous data on the effect of particle size, lamellarity and presence of cholesterol on the viscoelasticity of liposomes are already available in literature. Finally, the rheological characterization should be improved by adding for instance information on the thixotropic behavior of the different systems in study. Finally, authors should also provide at least some representative amplitude graphics showing the linear viscoelastic region selected for the frequency test.
Minor comments:
- -Materials section: this section is incomplete. Some information on lipid used and equipment are indeed missing;
- - Tables should be re-organized for sake of clarity, for example adding a table on the quali-quantitative composition of prepared liposomes (please specify in the table if the component ratio is expressed as molar or weight ratio);
- - Please uniform the abbreviations for unilamellar liposomes (SUV or RSV);
- - Oscillatory rheological measurements (part 2.4): Parameters used during the amplitude and frequency sweep (e.g., shear strain, angular frequency) are completely missing;
- - Results- part 3.1: in this section, only general, theorical principle of viscoelasticity are reported. Instead, data presented in the tables are not discussed at all or even mentioned;
- -The discussion, as already mentioned, is in general poor and also contains misleading information. As an example, supporting references are actually not in line with the results of the paper (e.g. ref 43). The mention of the references in the text should be checked also because of some incorrect number (e.g. at line 279 reference 10 is reference 49, indeed).
I have no particul comments on the level of English
Author Response
First of all, the authors would like to thank the Reviewer for taking the time and contributing their knowledge to improve their manuscript. We would like to thank Reviewer 1 for all the valuable comments and remarks.
You can see our answers and comments below, we hope you find them appropriate.

Reviewer 2 Report
The research conducted by Kállai-Szabó investigated the viscoelasticity of various liposomal dispersions and evaluate the impact of lipid concentration, the presence of cholesterol or 1,2-distearoyl-sn-glycero-3-phosphocholine (DSPC) and the gelling agents polyvinyl alcohol (PVA) or hydroxyethylcellulose (HEC) on the viscoelasticity of vesicular systems. They further studied the effect of two model drugs (phenyl salicylate and caffeine) on the viscoelastic behavior of liposomal system. The work provides information that contributes to our understanding of liposome. However, there are a few areas that could be further improved:
1. To enhance the clarity of your paper, I recommend starting with a clear statement of the purpose at the beginning. Additionally, briefly explain why understanding the viscoelastic properties of liposomal systems is crucial or how it contributes to the field. This will provide a strong introduction and better contextualize your study
2. The Introduction section appears to jump between different examples and applications without a clear structure. To improve readability, I suggest organizing the information in a more logical manner. For instance, discuss tears and their rheological properties first, followed by synovial fluids, and finally liposomal systems and their applications. This will provide a clearer flow of ideas.
3. Some sentences in the paper are long and complex, which makes the text difficult to read and comprehend. I recommend breaking them down into shorter sentences to improve readability. Additionally, clarify any ambiguous phrases or terminology to ensure a clear understanding for the readers.
4. Considering that the interpretation, preparation, and characteristic analysis of liposomes are commonly used methods, Figures 1-5 may be redundant. It would be more effective to focus on presenting the main findings and their implications. If the information in these figures is crucial to support your conclusions, then please retain them but consider making them more concise.
5. The Structure of the discussion section appears to be a collection of individual observations without a clear structure. I suggest organizing the discussion based on the main findings and their implications. For example, start by discussing the impact of multilamellar vesicles (MLVs) on viscoelasticity, followed by the effects of model drugs, lipid concentration, and size/lamellarity of liposomal samples. This will improve the coherence and flow of the discussion.
6. It would be helpful to provide explanations or interpretations for the observed results. Clarify the underlying mechanisms or factors that contribute to the changes in viscoelasticity. This will help the reader understand the significance of your findings and provide a deeper understanding of the topic.
Some sentences in the paper are long and complex, which makes the text difficult to read and comprehend. I recommend breaking them down into shorter sentences to improve readability. Additionally, clarify any ambiguous phrases or terminology to ensure a clear understanding for the readers.
Author Response
First, the authors would like to thank the Reviewer for taking the time and contributing their knowledge to improve their manuscript. We would like to thank Reviewer 2 for all the valuable comments and remarks.
You can see our answers and comments below, we hope you find them appropriate (word file).
